# Peroral Endoscopic Myotomy in the Management of Zenker’s Diverticulum: A Retrospective Multicenter Study

**DOI:** 10.3390/jcm10020187

**Published:** 2021-01-07

**Authors:** Aleksandra Budnicka, Władysław Januszewicz, Andrzej B. Białek, Michal Spychalski, Jaroslaw Reguła, Michal F. Kaminski

**Affiliations:** 1Department of Gastroenterological Oncology, The Maria Sklodowska-Curie National Research Institute of Oncology, 02-781 Warsaw, Poland; olabudnicka@wp.pl (A.B.); w.januszewicz@gmail.com (W.J.); jregula@coi.waw.pl (J.R.); 2Department of Gastroenterology, Hepatology and Oncology, Medical Center for Postgraduate Education, 02-781 Warsaw, Poland; 3Department of Gastroenterology, Pomeranian Medical University, 70-204 Szczecin, Poland; bialekab@gmail.com; 4Center of Bowel Treatment, University of Lodz, 95-060 Brzeziny, Poland; mspych80@gmail.com; 5Institute of Health and Society, University of Oslo, 1130 Oslo, Norway

**Keywords:** Cricopharyngeus myotomy, Kothari-Haber scoring system, peroral endoscopic myotomy, submucosal tunneling endoscopic septum division, Zenker’s diverticulum

## Abstract

Background: Peroral endoscopic myotomy (POEM) is an emerging technique in the treatment of Zenker’s diverticulum (ZD). This study aimed to analyze the feasibility of Zenker’s POEM (Z-POEM) in a multicenter setting and assess its performance using a validated Kothari-Haber Scoring System newly developed for symptom measurement in ZD. Materials and methods: This was a multicenter retrospective study involving three Polish tertiary referral endoscopic units. The data of consecutive patients with symptomatic ZD treated with Z-POEM in Poland between May 2019 and August 2020 were retrieved and analyzed. Primary outcome measures were technical success and clinical success rate (<3 points in Kothari-Haber Score at 2–3 months follow-up). Secondary outcome measures included procedures’ duration, length of hospital stay, and adverse events. Results: 22 patients with symptomatic ZD were included. The mean age was 67.6 (±10.7) years, and 14 (63.6%) were male. All but two patients were treatment naïve. The average size of the ZD was 30 mm (IQR, 24–40 mm). Technical success was achieved in all patients (100%), whereas clinical success was 90.9%. The average Kothari-Haber Score was 6.35 before treatment and has dropped to 0.65 after the treatment (*p* < 0.0001). The mean procedure time was 48.8 (±19.3) minutes, and the median length of hospital stay was 2 days (IQR, 2–3). Three patients (13.6%) had post-procedural emphysema, of which two were mild and self-resolving (9.1%), and one was moderate (4.5%) and complicated with laryngeal edema and prolonged intubation. Conclusions: This feasibility study suggests that Z-POEM is a highly effective and safe treatment for ZD, particularly among treatment-naïve patients. Comparative studies with other treatment modalities over longer follow-up are warranted.

## 1. Background

The Zenker’s diverticulum (ZD) is an acquired sac-like outpouching located dorsally at the pharyngoesophageal junction [1]. It is believed to originate from reduced compliance (opening) of the upper esophageal sphincter and increased intrabolus pressure [2]. In consequence, patients develop symptoms due to accumulation of food in the outpouching and difficult bolus transit through the uncompliant upper esophageal sphincter [2]. Patients can present with various symptoms, of which dysphagia is the most common. Other symptoms may include: regurgitation of undigested food, chronic cough, hoarseness, halitosis, throat clearing, choking, and (in severe cases) unintentional weight loss, malnutrition, and aspiration pneumonia [3]. Although multiple treatment options are available for symptomatic ZD, including open surgical cricopharyngeal myotomy, endoscopic stapled diverticulostomy, and flexible endoscopic diverticulotomy, all of these are considered suboptimal. Specifically, the open surgical approach has high clinical success rates; however, it is associated with a relatively high morbidity rate (10.5%) [4]. This is particularly important for elderly, comorbid patients with ZD [4]. Endoscopic stapled diverticulostomy is less invasive; however, associated with a relatively high recurrence rate of 11% [4]. Moreover, this technique cannot be applied in patients with reduced neck extension or mouth opening, which is common among elderly patients. Flexible endoscopic diverticulotomy is the least invasive approach, although limited by the incomplete myotomy, which usually leads to multiple treatment sessions and a relatively high recurrence rate of 15% [5]. A novel flexible endoscopic technique called peroral endoscopic myotomy (POEM) [6] has been developed to treat achalasia and was recently adopted for ZD treatment [7]. The advantage of Zenker’s POEM (Z-POEM) is that the submucosal tunneling allows for complete myotomy under direct visualization of the muscular septum, leading to a reduced number of treatment sessions and recurrences [8]. The evidence for Z-POEM’s efficacy is limited, and the previous studies did not use validated scales for the symptom measurement in ZD [9,10]. This study aimed to analyze the feasibility of the Z-POEM procedure in a multicenter setting and assess its performance using a novel Kothari-Haber Scoring System for symptom measurement in ZD.

## 2. Materials and Methods

This retrospective cohort study was conducted in three academic high-volume endoscopy centers in Poland between May 2019 and August 2020. Consecutive adult patients with symptomatic ZD with or without prior endoscopic or surgical treatment who underwent Z-POEM procedures were included. Each ZD diagnosis and measurement was made with either barium swallow or computed tomography and confirmed in direct endoscopic visualization. Relevant demographic, clinical, and procedural data were collected for each patient (e.g., symptoms, ZD size, procedural time, length of hospital stay, the occurrence of complications and adverse events, and follow-up data). The severity of adverse events was graded according to the American Society of Gastrointestinal Endoscopy (ASGE) lexicon [11]. The ZD-related symptoms were reported using the Kothari-Haber Scoring System (KHSS) immediately before the procedure and 2–3 months afterwards [3].

The KHSS is a novel validated tool used to assess the response to endoscopic myotomy for symptomatic ZD [3]. This scoring system is based on the most common clinical presentations seen in symptomatic ZD, including weight loss, dysphagia, oropharyngeal symptoms (halitosis, regurgitations), and respiratory symptoms (hoarseness or cough, pneumonia) with a combined score ranging from 0 (“no symptoms”) to a total score of 16 (“worst symptoms”), as presented in Table 1. The Institutional Review Board from the hosting institution (Postgraduate Medical Education in Warsaw) approved the study [91/PB/2019]. This study provides a foreground for an upcoming randomized clinical trial comparing Z-POEM with flexible endoscopic septotomy in patients with symptomatic ZD (the “ZIPPY” trial, NCT04514042).

### 2.1. Z-POEM Procedure

Patients were admitted within 24 h before the procedure after a fasting period of at least 6 h. A single intravenous dose of antibiotics as prophylaxis was administered before the procedure (e.g., amoxicillin and clavulanate, ceftriaxone, or ciprofloxacin). Procedures were performed using CO2 insufflation and under general anesthesia with endotracheal intubation. During endoscopy, the ZD was identified, and a mucosal bleb was created 1–2 cm proximal to the septum or at the septum using 5–10 mL of a mixed solution of 1% indigo carmine and normal saline. A 1.5 cm mucosal incision was performed along the septum’s long axis with a dual knife (dry-cut mode, 30 W, effect 2). The submucosal space was entered with the aid of the cap. Submucosal tunneling was performed in the septum direction using swift coagulation mode (effect 3, 40 W). After the tunneling, the septum was exposed. The dual knife (dry-cut mode, 30 W, effect 2) was also used to perform the septotomy. Myotomy was extended to the septum base reaching esophageal longitudinal muscle fibers delineating the completion of septum dissection or 0.5–1.0 cm below. The mucosotomy was securely closed using through-the-scope (TTS) clips. The Z-POEM procedure is presented in Appendix A.

Following the procedure, all patients were admitted for observation and kept nil per os overnight. Patients could resume a semiliquid diet on the first day after the procedure, followed by a soft diet for the next ten days if no adverse events occurred. A barium esophagram was performed in each case on the day after the procedure to exclude leakage. Patients were consulted via telephone for a follow-up visit after 2–3 months to assess the clinical response to the treatment or presence of any late adverse events and some of them were further followed-up at in-person visit 6 months or later after the treatment.

### 2.2. Study Outcomes

In this study, the two primary outcomes were the rate of technical success of the Z-POEM procedure, defined as completion of all procedural steps, and the procedure’s clinical success, defined as a complete resolution of dysphagia and complete or near-complete resolution of combined ZD-related symptoms (post-procedural KHSS < 3 points during the follow-up).

The secondary outcomes were the rate of procedure-related complications, adverse events, and hospital stay length.

### 2.3. Statistical Analysis

Quantitative variables were described as means and medians and standard deviations (SD) and interquartile ranges (IQRs), where appropriate. Categorical variables were presented as counts and percentages. The combined symptom scores before and after the treatment were compared using the Wilcoxon signed-rank test. A *p*-value of less than 0.05 was considered statistically significant. All analyses were performed using R Statistics version 3.4.3 (R Foundation for Statistical Computing, Vienna, Austria).

## 3. Results

Twenty-two patients with symptomatic ZD who underwent the Z-POEM procedure were included in the study, of which 14 were males (63.3%), with a mean age of 67.6 (±10.4) years. Two patients (9.1%) had previous endoscopic treatment with argon plasma coagulation septotomy. All patients presented with various degree of dysphagia, and the concurrent symptoms included: regurgitations (19/22, 86.4%), hoarseness (14/22, 63.6%), cough (12/22, 54.5%), halitosis (8/22, 36.4%), weight loss (7/22, 31.8%) and aspirational pneumonia (3/22, 13.6%). The median ZD length was 30mm (IQR 20–40 mm). Patient baseline characteristics are presented in Table 2.

Complete peroral endoscopic myotomy was technically successful in all 22 patients (100%). The mean procedural time (measured from the beginning of the initial submucosal injection to the last clip’s placement during the mucosal defect closure) was 48.8 (±19.4) minutes, and the median hospital stay was 2 days (IQR 2–3). The mucosal entry to the submucosal tunnel was sealed with a median number of 5 clips (IQR 4–5).

Intra-procedural complications occurred in three patients (13.6%), including:Mild: two cases of transient and asymptomatic subcutaneous emphysema of the neckModerate: a single case of an extensive subcutaneous and intramuscular emphysema of the neck, chest, and abdomen followed by laryngeal edema requiring prolonged intubation and intensive care unit monitoring. This likely occurred due to erroneous simultaneous CO_2_ and air insufflation.

Post-procedural adverse events occurred in three individuals (13.6%). These included three cases of a sore throat (mild). No severe or fatal events were observed.

Clinical success was achieved in nearly all patients (20/22, 90.9%). The mean KHSS has dropped from 6.4 (±2.1) points (pre-treatment) to 0.6 (±1.2) points after the treatment (*p* < 0.001), as presented in Figure 1. All patients stabilized or gained weight after the treatment. Two patients (9.1%) had persistent dysphagia with accompanying respiratory symptoms; however significantly milder than before the treatment. One of the abovementioned patients had recurrent ZD after previous treatment, and his septum was still “standing” after complete myotomy, most likely due to severe fibrosis. The symptoms wholly resolved (KHSS 0 points) after subsequent septotomy of the fibrotic septum. The reason for incomplete response in a second patient with persistent dysphagia is unclear. The patient had significant symptomatic improvement (8 to 3 points, see Figure 1) and did not decide to undergo additional treatment.

None of the patients had an increase in symptom severity over a median follow-up of 266 days (IQR 213–306 days). Procedural features are summarized in Table 3.

## 4. Discussion

This multicenter study showed that Z-POEM is a feasible treatment option for symptomatic ZD, with 100% technical and 91% clinical success rates, measured with a validated KHSS tool. Z-POEM was effective in relieving dysphagia and other ZD-related symptoms, such as regurgitations, hoarseness, cough, or halitosis. The short-term clinical success rate among treatment-naïve patients was 95%. None of the patients had a recurrence over the median 266 days follow-up. The rate of adverse events was low, with all but one being mild according to the ASGE guidelines. Despite a short follow-up time, this study confirms that a single Z-POEM procedure is highly effective and suitable for a head-to-head comparison with flexible endoscopic diverticulotomy in a randomized trial (the “ZIPPY” trial, NCT04514042).

The efficacy of the Z-POEM procedure within our study is comparable to that of previous reports [9,10,12]. Although measured with different symptom scores, the technical and clinical success rates of the earlier studies were ranging between 97% to 100% and 86% to 95%, respectively [9,10,12]. We have shown that the myotomy alone (without cutting mucosal parts of the septum) is capable of resolving a broad spectrum of ZD-related symptoms, beyond dysphagia only. This finding confirms that myotomy of the upper esophageal sphincter plays a central role in ZD treatment and resolves the concern that remnant mucosal excess may cause persistent regurgitation [13]. In our experience, remnant mucosal excess may cause persistent symptoms only in the case of recurrent ZD in which fibrotic mucosa prevents its collapse after myotomy.

The rate of adverse events in all reported studies, including ours, is below 10%, of which less than 5% are moderate or severe [9,10,12]. The average procedure time was 48 to 52 min in all but one study, in which a simplified technique for short septum ZD was used [9]. This compares unfavorably with the average time of conventional flexible endoscopic diverticulotomy, which lasts on average 15 min and can be performed without general anesthesia [5]. This further calls for a direct performance comparison of Z-POEM with flexible endoscopic diverticulotomy.

This study’s main strength was that the treatment response assessment was made with a validated scoring system (KHSS) dedicated to symptom assessment in ZD. All previous studies used scoring systems focused exclusively on the severity of dysphagia, which is not fully representative of ZD’s symptom burden. An additional advantage of this study was the inclusion of all and consecutive patients who underwent Z-POEM in Poland, representing its real-world performance at the introduction phase of this treatment. This study’s main limitations included retrospective design, which caused a degree of missing data, a relatively small sample size, and short-term follow-up.

In conclusion, this feasibility study shows that Z-POEM is a highly effective and safe treatment modality for ZD, particularly among treatment-naïve patients. Comparative studies with other treatment options over a longer follow-up time are highly warranted.

## Figures and Tables

**Figure 1 jcm-10-00187-f001:**
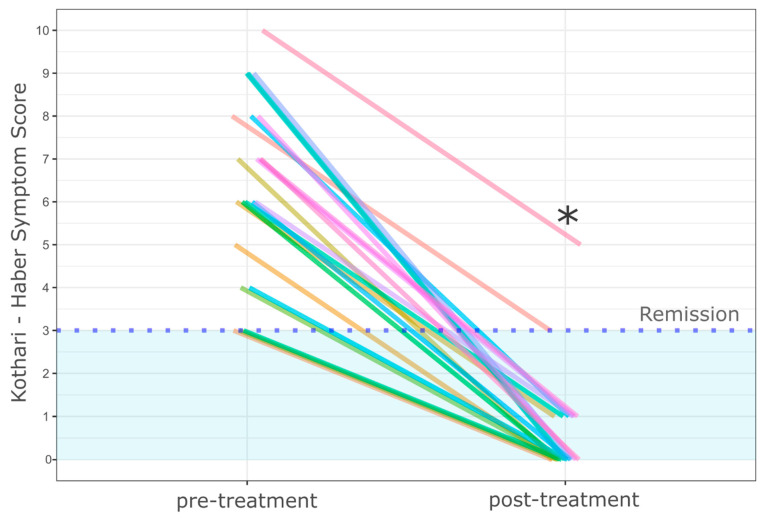
The symptom severity assessed with the Kothari-Haber Scoring System before and after Zenker’s peroral endoscopic myotomy (Z-POEM) treatment. * The patient had recurrent ZD after previous treatment and his septum did not collapse after complete Z-POEM, most likely due to severe fibrosis. The symptoms completely resolved (KHSS 0 points) after subsequent septotomy of the fibrotic mucosal septum. Each line represents different patient.

**Table 1 jcm-10-00187-t001:** Kothari-Haber Scoring System for Zenker’s Diverticulum [3].

Symptom	Score
Weight loss (over one year)	
none	0
1–10 lbs (0.45–4.54 kg)	1
11–20 lbs (4.99–9.07 kg)	2
>20 lbs (>9.07 kg)	3
Dysphagia	
none	0
solids	1
semi-solids	2
solid and liquids	3
Oropharyngeal Symptoms	
Regurgitation	
none	0
occasional	1
daily	2
Halitosis	
none	0
occasional	1
daily	2
Respiratory Symptoms	
Cough	
none	0
occasional	1
daily	2
Hoarseness	
none	0
occasional	1
daily	2
Pneumonia	
No	0
Yes	2

**Table 2 jcm-10-00187-t002:** Patients’ characteristics.

Characteristic	Value
Age, mean ± SD, years	67.55 ± 10.7
Male gender—*n* (%)	14 (63.3%)
BMI, median (IQR), kg/m^2^	24.5 (23.9–27.0)
Prior endoscopic treatment—*n* (%)	2 (9.1%)
Size of ZD, median (IQR), mm	30 (20–40)
Presenting symptoms—*n* (%)	
Dysphagia	22 (100%)
Regurgitations	19 (86.4%)
Hoarseness	14 (63.3%)
Cough	12 (54.5%)
Halitosis	8 (36.4%)
Weight loss	7 (31.8%)
Aspirational pneumonia	3 (13.6%)

IQR: interquartile range; SD: standard deviation; ZD: Zenker’s diverticulum.

**Table 3 jcm-10-00187-t003:** Technical and clinical outcomes of the Zenker’s peroral endoscopic myotomy technique.

Outcomes	Value
Technical success—*n* (%)	22 (100%)
Clinical success—*n* (%)	20 (90.9%)
Procedural time, mean ± SD, min	48.8 (±19.4)
Days of hospitalization, median (IQR)	2 (2–3)
Intraprocedural complications—*n* (%) Mild: subcutaneous emphysema of the neck (2 patients) Moderate: Extensive subcutaneous and intramuscular emphysema of the neck, chest, and abdomen (1 patient)	3 (13.6%)
Postprocedural adverse events—*n* (%) Sore throat (3)	3 (13.6%)
Pre-procedure Kothari-Haber System Scoring, mean (±SD)	6.4 (±2.1)
Post-procedure Kothari-Haber System Scoring, mean (±SD)	0.6 (±1.2)

## Data Availability

The data presented in this study are available on request from the corresponding author. The data are not publicly available due to restrictions on privacy.

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
