# Peer review of "Peroral Endoscopic Myotomy in the Management of Zenker’s Diverticulum: A Retrospective Multicenter Study"

_jcm, 2021, doi:10.3390/jcm10020187_

Round 1

Reviewer 1 Report

I congratulate the authors for their nice multicenter pilot study.

The paper is well written, just some minor formatting issues detected.

The Kothari-Habor Score was well presented and used to objectively measure success.

Author Response

Thank you very much for the review.

Reviewer 2 Report

The authors present a three center retrospective review of their experience with Z-POEM. This was done aparrently as a preliminary to a randomized prospective study between Z-POEM and standard endoscopic Zenkers diverticulectomy. The procedures went well and there was a 90% symptoms success rate at 8 months. 2 patients had fairly early recurrence and one went reintervention. A strength of this report is the use of the KHSS symptom score which is specific for this disease

  • Please describe better your follow up for these patients, In methods you state that they were contacted at 2-3 months after the procedure and yet you report a mean followup between 7 and 9 months. How was the followup done, by phone? all patients could be reached? or are we only looking at the data of the ones answering the phone at 8 months?
  • 2 patients failed, you describe how one was treated, please add more information regarding the second patient, why did they ffail? persistant symptoms or recurrent? When was recurrence? how was it dealt with?
  • It seems that the value of all of the extra work of bilateral submucosal tunneling is extending the myotomy onto the esophageal wall, thereby making sure all of the cricopharyngeous is divided. As you mention the end result is sometimes complicated by a neo-septum of mucosa. wouldn't it be quicker, easier and even better, to do a septotomy to the end of the pouch, then tunnel a centimeter or so down the esophagus to extend the myotomy - and then close the defect transversely with clips?  just a thought and something you might mention in the discussion
  • \thanks 

Author Response

Comment 1:

Please describe better your follow up for these patients, In methods you state that they were contacted at 2-3 months after the procedure and yet you report a mean followup between 7 and 9 months. How was the followup done, by phone? all patients could be reached? or are we only looking at the data of the ones answering the phone at 8 months?

Response1:

Thank you for this comment. The follow-up was one by phone at 2-3 months and some patients were followed-up in person at 6 months or later (due to COVID19 pndemic some of the visits were cancelled or rescheduled). This has now been revised in the methods section of the manuscript.

Comment 2:

2 patients failed, you describe how one was treated, please add more information regarding the second patient, why did they ffail? persistant symptoms or recurrent? When was recurrence? how was it dealt with?

Response 2:

These were persistant symptoms. No patient had recurrence following initial clinical success. We have no clear explanation for the second failure (this was one of our initial cases so we could have overlooked potential reason for the failure). The patient had a post-treatment score of 3 (K-H score) which was considerable improvement from the initial score of 8. The patient was OK with the result of the treatment so we left the option for futher treatment at his discretion. We have added clarifications to the results section.

Comment 3:

It seems that the value of all of the extra work of bilateral submucosal tunneling is extending the myotomy onto the esophageal wall, thereby making sure all of the cricopharyngeous is divided. As you mention the end result is sometimes complicated by a neo-septum of mucosa. wouldn't it be quicker, easier and even better, to do a septotomy to the end of the pouch, then tunnel a centimeter or so down the esophagus to extend the myotomy - and then close the defect transversely with clips? just a thought and something you might mention in the discussion

Response 3:

This may indeed be a very good idea. The two potential problems are that after septotomy and extended myotomy a. the visualization of the plains and layers at the phase of extended myotomy could be more difficult and b. closure of the defect after long septotomy could be difficult. Nevertheless it is for sure worth being tried in an experimental setting. No comments to the manuscript.

Reviewer 3 Report

This paper presents the results of a newer interventional technique to address Zenker’s diverticula. 22 consecutive patients underwent peroral myotomy with very good technical and clinical success. This paper is fluently written, has an impact on treatment of ZD and is of timely relevance. I just have a few remarks and questions:

Line 50: I wouldn’t say the morbidity is high (actually quite the contrary), it depends on the technique used, resection of the sac or diverticulopexy. Please provide some numbers

Line 70: do you refer to size of ZD or rather the opening in the triangle? If former, how is it measured or standardized (CT vs barium swallow)?

Does size of sac play a role in indication for Z-POEM (bulky, big intrathoracic ZD also amenable?)

Lines176ff: as stated above, this experience is not novel. In open procedures, diverticulopexy has the same results with less morbidity. However, size of the sac has an importance (mostly if very big and fixed).

Author Response

Line 50: I wouldn’t say the morbidity is high (actually quite the contrary), it depends on the technique used, resection of the sac or diverticulopexy. Please provide some numbers

Comment: Thank you for the comment, we provided numbers in the text.

Line 70: do you refer to size of ZD or rather the opening in the triangle? If former, how is it measured or standardized (CT vs barium swallow)?

Comment: The former. It was measured on barium swallow or CT. We added the information to the methods section.

Does size of sac play a role in indication for Z-POEM (bulky, big intrathoracic ZD also amenable?)

Comment: We don't think that very big sac is a contraindication to Z-POEM. The biggest sac in this series was 9 cm in size! We think that proximity to large vessels/vital organs with big vessels could be a limitation.

Lines176ff: as stated above, this experience is not novel. In open procedures, diverticulopexy has the same results with less morbidity. However, size of the sac has an importance (mostly if very big and fixed).